# Mitochondrial OXPHOS Biogenesis: Co-Regulation of Protein Synthesis, Import, and Assembly Pathways

**DOI:** 10.3390/ijms21113820

**Published:** 2020-05-28

**Authors:** Jia Xin Tang, Kyle Thompson, Robert W. Taylor, Monika Oláhová

**Affiliations:** 1Wellcome Centre for Mitochondrial Research, Newcastle University, Newcastle upon Tyne NE2 4HH, UK; j.x.tang2@newcastle.ac.uk (J.X.T.); kyle.thompson@newcastle.ac.uk (K.T.); robert.taylor@newcastle.ac.uk (R.W.T.); 2Newcastle University Translational and Clinical Research Institute, Newcastle University, Newcastle upon Tyne NE2 4HH, UK

**Keywords:** OXPHOS biogenesis, mitochondrial gene expression, mitochondrial import, OXPHOS assembly factors, mitochondrial ACP, LYRM proteins

## Abstract

The assembly of mitochondrial oxidative phosphorylation (OXPHOS) complexes is an intricate process, which—given their dual-genetic control—requires tight co-regulation of two evolutionarily distinct gene expression machineries. Moreover, fine-tuning protein synthesis to the nascent assembly of OXPHOS complexes requires regulatory mechanisms such as translational plasticity and translational activators that can coordinate mitochondrial translation with the import of nuclear-encoded mitochondrial proteins. The intricacy of OXPHOS complex biogenesis is further evidenced by the requirement of many tightly orchestrated steps and ancillary factors. Early-stage ancillary chaperones have essential roles in coordinating OXPHOS assembly, whilst late-stage assembly factors—also known as the LYRM (leucine–tyrosine–arginine motif) proteins—together with the mitochondrial acyl carrier protein (ACP)—regulate the incorporation and activation of late-incorporating OXPHOS subunits and/or co-factors. In this review, we describe recent discoveries providing insights into the mechanisms required for optimal OXPHOS biogenesis, including the coordination of mitochondrial gene expression with the availability of nuclear-encoded factors entering via mitochondrial protein import systems.

## 1. Introduction

With the exception of erythrocytes, mitochondria are ubiquitous organelles within eukaryotic cells owing to their essential function of supplying cellular energy in the form of adenosine triphosphate (ATP) via oxidative phosphorylation (OXPHOS) carried out by the electron transport chain (ETC) and ATP synthase. The failure of mitochondria to achieve this is associated with a wide spectrum of genetic disorders affecting both children and adults with a minimum prevalence of one in 4300 [1]. In humans, the OXPHOS system comprises five multimeric protein complexes embedded in the inner mitochondrial membrane (IMM). The respiratory chain complexes (Complexes I–IV (CI–IV)) are responsible for the transfer of electrons released from the reduced forms of nicotinamide adenine dinucleotide (NADH) and flavin adenine dinucleotide (FADH_2_) reducing equivalents to oxygen, the final electron acceptor, through a series of exergonic redox reactions that contribute to the generation of the electrochemical gradient across the IMM. The proton gradient drives the translocation of protons from intermembrane space (IMS) back into the matrix via the ATP synthase (Complex V (CV)) that catalyzes the conversion of ADP to ATP.

The spectrum of pathological conditions that can be attributed to mutations in individual OXPHOS subunits and assembly proteins is often characterized by a complex-specific biochemical defect. The importance of a large number of these proteins in human pathology was previously described [2,3,4], and this review focuses on the fundamental role of mitochondria in bioenergetics, covering major molecular process underpinning the architecture and assembly of the OXPHOS system that involves mito-nuclear co-regulation on multiple levels: the coordination of distinct gene expression programs, protein import and assembly pathways.

### Coupling of Catabolic Pathways and the OXPHOS System

Mitochondria catabolize nutrients to fulfil their role in cellular bioenergetics. In order to do this, mitochondria heavily rely on acetyl coenzyme A (acetyl-CoA), a product of many catabolic pathways, that serves as a primary substrate for the ETC. For instance, pyruvate, a product of glycolysis in the cytosol, is transported into mitochondria and converted into acetyl-CoA in a reaction catalyzed by the mitochondrial pyruvate dehydrogenase complex (PDHC). Acetyl-CoA enters a series of enzyme-catalyzed reactions in the tricarboxylic acid (TCA) cycle producing NADH and FADH_2_ reducing equivalents (Figure 1) [5,6]. NADH and FADH_2_ feed CI and CII of the ETC, respectively, generating an electrochemical gradient across the IMM, thus contributing to the proton-motive force which drives the ATP synthesis (Figure 1). Fatty acids are also a vital source of cellular energy metabolized in mitochondria by the fatty acid β-oxidation (FAO) pathway. Cytosolic fatty acids are activated by acyl-CoA synthases, converted into fatty acyl-CoA esters, and transported into mitochondria via a designated carnitine shuttle system. Within mitochondria, fatty acyl-CoAs enter the FAO spiral resulting in (i) chain-shortening of the entry product and its recycling via the FAO pathway, (ii) generation of acetyl-CoA that contributes to the overall mitochondrial pool of acetyl-CoA used in the TCA cycle, and (iii) the production of reducing equivalents (NADH and FADH_2_) coupling FAO and the OXPHOS system (Figure 1). Finally, amino-acid degradation also contributes to the production of pyruvate, acetyl-CoA, or metabolic intermediates that are oxidized in the TCA cycle (Figure 1). It is evident that mitochondria accomplish many vital functions in cellular metabolism as the hosting centers for the TCA cycle and FAO pathway [5,6]. However, in addition to this, the multifaceted roles of mitochondria in iron–sulfur (Fe–S) cluster biogenesis, apoptosis, calcium handling, and reactive oxygen species signaling are key in the maintenance of balanced cellular homeostasis [7,8,9,10].

## 2. Co-Regulation of Mitochondrial and Nuclear Gene Expression

OXPHOS function is under the dual genetic control of both the nuclear and mitochondrial genomes. The assembly process of each multimeric complex is an intricate process involving many nuclear and mitochondrial factors. Nuclear-encoded gene products of the OXPHOS complexes are transcribed and translated in the cytosol before being imported into the organelle via mitochondrial translocase machineries. In contrast, the small circular mitochondrial DNA (mtDNA) (16,569 bp) encodes 37 gene products, including two ribosomal RNAs (rRNAs) and 22 transfer RNAs (tRNAs) necessary for mitochondrial translation and 13 OXPHOS polypeptides, which are locally transcribed and translated within the mitochondria [11]. It is estimated that about ~250 nuclear-encoded proteins are important for optimal maintenance and expression of the mitochondrial genome, with a total of around 1000 proteins in yeast [12] and up to 1400 in humans—all of which are necessary for mitochondrial function [13,14]. Hence, a tightly regulated coordination between nuclear and mitochondrial gene expression machineries is essential. This is to ensure correct and efficient synthesis, transport, localization, and assembly of the cytosolic and mitochondrial proteins to constitute the OXPHOS system in the mitochondria despite originating from distinct cellular compartments.

### 2.1. Mitochondrial Transcription

Mitochondrial gene expression follows the central dogma of nuclear gene expression, which requires transcription of DNA to messenger RNA (mRNA) followed by translation of mRNA into an amino-acid sequence of a protein. However, there are several differences that cause mitochondrial gene expression to differ from nuclear gene transcription and translation. For instance, the mitochondrial genome is made up of heavy and light strands and transcribed from both strands as long polycistronic transcripts [15].

The transcription of mtDNA relies entirely on nuclear-encoded factors [16]. Mitochondrial RNA polymerase (POLRMT), the main player in this process, is a DNA-dependent RNA polymerase which is highly similar to phage T7 polymerase [17,18,19]. Additional factors need to be recruited for the initiation of mtDNA transcription, namely, mitochondrial transcription factor A (TFAM) and mitochondrial transcription factor B2 (TFB2M) (for a review, see Reference [20]). Following dissociation of TFAM and TFB2M initiation factors, mitochondrial elongation factor (TEFM) associates with POLRMT to form a sliding clamp to ease elongation of mitochondrial RNA by reshuffling the topology of the DNA template [21]. The termination of mtDNA transcription on the light strand was uncovered with the identification of mitochondrial transcription termination factor 1 (MTERF1) that stops progression of POLRMT at the 3′ end of mtRNA sequences [22,23,24]. Although the mode of termination for the heavy strand remains elusive, it was suggested that partial DNA unwinding, likely by base-flipping, may induce termination of transcription on the heavy strand [25].

Mitochondrial transcription generates three main polycistronic transcripts, which require further processing for maturation of individual RNAs, a process that occurs in discrete foci called mitochondrial RNA granules [26]. The “tRNA punctuation model” was put forward in 1981, proposing that the sequence of the mitochondrial polycistronic transcripts is punctuated by tRNAs [27]. Cleavage of the 5′ and 3′ end of tRNAs by Ribonuclease P (RNaseP) and Ribonuclease Z (RNaseZ) complexes is required to release individual mRNAs and rRNAs from the long polycistronic transcripts. Nonetheless, some mRNAs are not flanked by tRNA molecules and are, therefore, processed through non-canonical pathways for maturation (for a review, see Reference [28]). Unlike nuclear mRNA, mt-mRNA molecules, are devoid of 5′ caps, 5′ and 3′ untranslated regions (UTRs), and introns. Processed mt-RNAs undergo different modifications for mRNA maturation into functional RNA molecules for mitochondrial translation. The two most common mt-RNA post-transcriptional modifications are 3′ polyadenylation, which is essential for the completion of the stop codon UAA for seven transcripts, and the addition of the conserved CCA sequence to the 3′ end of mt-tRNAs for cognate amino-acid attachment [29,30,31,32].

### 2.2. Mitochondrial Translation

Mitochondria possess their own ribosomes, referred to as mitoribosomes. Despite the distinct characteristics of mt-mRNAs, the translation process in mitochondria is functionally similar to cytosolic translation. The structure of the mitoribosome is similar to its bacterial and cytoplasmic counterparts in that it consists of a large subunit and a small subunit, but it differs in size and has a higher proportion of protein-to-rRNA ratio, owing to evolutionary events involving the acquisition of novel protein subunits followed by reduction of the mitochondrial genome [33,34,35]. The human 55S mitoribosome is made up of (i) a 28S small subunit (mt-SSU) consisting of 12S rRNA and 30 mitochondrial ribosomal proteins (MRPs), and (ii) a 39S large subunit (mt-LSU) consisting of 16S rRNA and 52 MRPs excluding mL56 which was shown to function as a serine beta-lactamase (LACTB) protein to modulate mitochondrial lipid metabolism [36,37,38]. It is notable that all proteins involved in mitochondrial translation including those required for mitoribosome biogenesis and assembly are nuclear-encoded, again emphasizing the importance of effective co-ordination between nuclear and mitochondrial gene expression to facilitate mitochondrial protein synthesis [39].

#### Differences between Cytosolic and Mitochondrial Translation

While the cytosolic translation is initiated by the binding of initiation factors to the 5′ cap and poly(A) binding protein to the 3′ poly(A) tail, mitochondrial mRNAs lacking those characteristics initiate translation through the binding of mitochondrial translation initiation factors to the start codon, AUG or AUA, directly at the codon recognition site in mt-SSU [37,39]. Cytosolic translation requires the formation of pre-initiation complex through joining of the 40S small subunit to the initiation factors to form a scanning complex to identify the start codon, AUG, in the 5′ UTR [40]. Another difference in mt-mRNA translation initiation is that the start codon and amino-acid methionine share the same trinucleotide sequence. Methionyl-tRNA formyltransferase, therefore, plays an important role in adding formyl group to tRNA^Met^ to signal initiation of mitochondrial translation, allowing association of the large subunit for monosome assembly [41,42]. The elongation of peptides takes place in the peptidyl transferase center in the mt-LSU, which then moves the nascent peptide chain along by three nucleotides for translation of the subsequent codon on the mRNA sequence. In most cases, termination occurs when stop codons (UAA/UAG) are recognized by the mitochondrial translation release factor 1 (MTRF1) that facilitates hydrolysis of the ester bond between peptidyl-tRNA and the terminal amino acid of the nascent polypeptide chain [43]. Once mitochondrial translation is terminated, the mitochondrial ribosome release factor (MRRF) and mitochondrial elongation factor (mtEFG2) facilitate the recycling of the mitoribosome and release uncharged tRNA by binding to mitoribosomes with a deacylated tRNA in the peptidyl site [44,45].

### 2.3. Coordination of Distinct Translational Programs Facilitate Stochiometric OXPHOS Subunit Synthesis

It is evident that synchronization of two physically separated gene expression systems is necessary for production of dual-origin OXPHOS complexes. Determining the mechanisms that regulate these events remains to be discovered. However, it is clear that these processes not only require coordinated production of OXPHOS subunits and chaperones, but also efficient synthesis of numerous nuclear-encoded proteins involved in every stage of mtDNA maintenance and mitochondrial gene expression, assembly factors, and translational regulators to aid OXPHOS protein synthesis.

Mitoribosome biogenesis and assembly are of ultimate importance to ensure effective mitochondrial translation. Despite detailed characterization of the human mitoribosome structure attributed to cryogenic electron microscopy (cryo-EM) studies [36], the order of mitoribosome assembly is still not fully understood. Recently, Bogenhagen and colleagues modeled the assembly of the mitoribosome by utilizing pulse SILAC (Stable isotope labeling by amino acids in cell culture) of MRPs in HeLa cells [46]. Interestingly, many MRPs were found to be synthesized in excess, which is likely to compensate for the complex and lengthy process of mitochondrial ribosome assembly that was estimated to take two to three hours before achieving effective translation of mitochondrial proteins [46]. Furthermore, early- and late-binding MRPs were identified and differentiated into two groups through mass spectrometry (MS) [46]. These data suggest that assemblies of both mt-LSU and mt-SSU begin with the association of early-binding proteins with 12S and 16S rRNAs in clustered groups. The subgroups of early-binding MRPs contribute to distinct parts of the ribosomal subunit. Late-binding MRPs rely more on protein–protein interactions with the pre-assembled early-binding complexes rather than direct association with the 12S or 16S rRNA [46]. Additionally, the mitochondrial assembly of ribosomal large subunit protein 1, MALSU1 (C7orf30), was identified by cryo-EM as a late-assembly intermediate required for the biogenesis of the mt-LSU [47]. The mitoribosomes are anchored to the IMM by the mitochondrial large subunit L45 protein (mL45) to increase proximity for the insertion of hydrophobic mitochondrial proteins by the mitochondrial insertase OXA1L [46,48,49]. These findings collectively proposed the involvement of a number of nuclear-encoded mitoribosome assembly factors required for mitoribosome biogenesis, in addition to the excess synthesis of numerous MRPs—likely to allow for sufficient supply of subunits to aid mitoribosome biogenesis.

A study investigating the potential interlinkage between the nuclear and mitochondrial gene expression programs revealed that translation of both nuclear- and mtDNA-encoded OXPHOS transcripts were elevated when *Saccharomyces cerevisiae* were shifted from glucose-containing to glycerol-containing growth medium [50]. There is, therefore, even more emphasis on the communication between nuclear and mitochondrial gene expression for the maintenance of normal OXPHOS function under varying physiological conditions. Yeast can grow anaerobically on glucose but requires OXPHOS to meet cellular energy demands on glycerol. Ribosome profiling identified a redistribution of the translational efficiency of cytosolic and mitochondrial ribosomes in adaptation to the nutrient shift, suggesting a strong coordination between the two gene expression systems in regulating OXPHOS protein expression in response to nutrient status [50]. However, it appears that this a unidirectional relationship, as the regulatory changes in cytosolic translation following nutrient shift were maintained even when mitochondrial translation was inhibited, while mitochondrial gene expression was differentially affected by drug inhibition of cytosolic translation independent of the nutrient shift. The rate of increase in mitochondrial translation was also found to be slower compared to cytosolic translation, showing that the two gene expression systems are not adjusted concomitantly in response to physiological status [50].

## 3. Coupling of Mitochondrial Protein Import and OXPHOS Biogenesis

### 3.1. Nuclear-Encoded Mitochondrial Protein Import Systems

As mentioned previously, all nuclear-encoded mitochondrial proteins are translated on cytosolic ribosomes and need to be imported into mitochondria by specific protein translocases. There are five main import pathways that are described in mitochondria with each pathway specific to a subsection of precursor proteins depending on their structure and which mitochondrial compartment they are to be directed to. These are briefly outlined below, but more in-depth reviews of mitochondrial protein import can be found in References [51,52,53,54].

The vast majority of precursor proteins are imported by initial translocation across the outer mitochondrial membrane (OMM) by the TOM (translocase of outer membrane) complex. The exception are outer membrane proteins that have many alpha helices, which are recognized by TOM receptors but do not pass through the channel of the translocase; instead, they are imported by the mitochondrial import machinery (MIM). Outer membrane proteins that form beta-barrels are targeted to the SAM (sorting and assembly machinery) complex after passing through the TOM complex. Inter membrane space (IMS) proteins that require oxidative folding are directed through the MIA (mitochondrial IMS assembly) pathway. Hydrophobic precursor carrier proteins destined for the IMM are targeted through the translocase of the inner mitochondrial membrane 22 (TIM22) complex. The largest proportion of mitochondrial proteins have an N-terminal targeting sequence [55] and are passed through the TOM complex to the TIM23 complex, and they include proteins targeted to the mitochondrial matrix and the IMM. This is how the vast majority of OXPHOS components are imported, and this is the pathway discussed in more detail.

TIM23 is the presequence translocase of the IMM and either allows lateral insertion of inner membrane proteins or, in association with the PAM (presequence translocase-associated motor) complex, allows translocation of proteins into the mitochondrial matrix. The N-terminal presequence on these precursor proteins is positively charged, and this charge drives the translocation due to the mitochondrial membrane potential produced by the activity of the ETC. Translocation to the matrix is also ATP-dependent, as the activity of mtHsp70 (mitochondrial heat-shock protein 70), the central subunit of the PAM complex, is required for transport. Most OXPHOS subunits are imported by the TIM23 complex, many of which are inserted laterally directly from the TIM23 complex. Some IMM proteins, however, are imported into the matrix through TIM23 and PAM and are then inserted into the IMM by Oxa1 (oxidase assembly protein 1).

### 3.2. The Function of the Oxa1 Insertase in OXPHOS Assembly

Oxa1 is a member of the YidC/Alb3/Oxa1 membrane protein insertase family [56]. The yeast protein Oxa1p is the most studied, and it was first identified as an important factor for the assembly of CIV [57], while it was subsequently shown to be important for the assembly of CV [58]. Oxa1p is known to have a direct role in the insertion of several nuclear-encoded IMM proteins [59] including Oxa1p itself [60], and it has an indirect effect on many more IMM proteins, including several metabolite transporters [61], since it is crucial for the biogenesis of the Tim18-Sdh3 module of the TIM22 carrier translocase in yeast [62]. Oxa1p and its human homolog OXA1L are also required for the co-translational insertion of most mtDNA-encoded proteins including Atp6p, Atp9p, Cox1p, Cox2p, Cox3p, and Cytb in yeast [59,60,63,64], and OXA1L interacts with at least nine mtDNA-encoded subunits in humans (COX1, COX2, COX3, ND1, ND2, ND3, ND4, ND5 and ATP6) [65]. Expression of OXA1L partially rescued the phenotype of impaired cytochrome *c* oxidase (COX) assembly in an Oxa1p null strain of *Saccharomyces cerevisiae*, suggesting that OXA1L likely performs similar roles to the yeast Oxa1p in human cells [66]. Indeed, both Oxa1p [67,68] and OXA1L [69] were shown to interact with the mitoribosome, suggesting further conservation of function and highlighting the importance of close proximity of the translation and insertion machinery.

### 3.3. Importance of Close Proximity of Import, Insertase, and Assembly Machinery

Close proximity of the import translocase machinery to the sites of respiratory complex subunit insertion and assembly is important, since components encoded by both the mtDNA and nuclear genome need to be coordinated and assembled concurrently. Electron microscopy studies in yeast showed that early stages of CIII and CIV assembly are enriched at the inner boundary membrane close to the import machinery, whereas the mature complexes are more enriched in the cristae membrane [70]. Similar studies showed that Oxa1p can move between these sections of the IMM and is enriched in the inner boundary membrane in fermentable growth conditions; however, under respiratory growth conditions, it is enriched in the cristae membrane [71], suggesting that it changes its predominant location based on the availability of substrates. Equivalent data on the distribution of assembly intermediates and OXA1L in human mitochondria are so far lacking. More direct evidence for the importance of close proximity between the protein import machinery, particularly the TIM23 complex, respiratory chain assembly, and the OXA1L insertase, comes from study of the MITRAC complex.

The MITRAC complexes (mitochondrial translation regulation assembly intermediate of cytochrome *c* oxidase) were first described as assembly intermediates of CIV and include subunits such as MITRAC12 (COA3), C12orf62 (COX14), MITRAC15 (COA1) and TIM21 [72]. The association of the TIM23 complex subunit, TIM21, with MITRAC complexes demonstrates the coordination of import and assembly of nuclear-encoded OXPHOS components [72]. Subsequently, it was shown that MITRAC15 is more important for CI assembly as it interacts with the newly synthesized mtDNA-encoded subunit ND2, as well as the CI assembly factor, Acyl-CoA dehydrogenase family member (ACAD9), to promote early assembly of the ND2/P_P-b_ module of CI (Figure 2A) [73]. TIM21 was shown to interact with MITRAC12 and several other early CIV assembled proteins including nuclear-encoded COX4 and mtDNA-encoded COX1, COX2, and COX3 [72]. In addition, TIM21 also interacts with MITRAC15 and ACAD9 in the ND2/P_P_-b module [73] and other mtDNA-encoded proteins of CI (ND2, ND4, ND5) and CIII (CYTB) [72], suggesting a more general role for TIM21 in respiratory complex assembly. TIM21 does not strongly interact with MITRAC7, which is a COXI-specific chaperone that is present in later-stage MITRAC complexes and acts downstream of TIM21, incorporating early nuclear-encoded subunits into assembly intermediates of CIV [74]. FLAG immunoisolation in both MITRAC15^FLAG^ and C12orf62^FLAG^ expressing cell lines also identified mitochondrial ribosomal proteins and OXA1L as interacting partners [73], while affinity purification MS analysis of proteins interacting with OXA1L^FLAG^ identified MITRAC12, ACAD9, and TIM21 among the proteins with a greater than three-fold enrichment in OXA1L^FLAG^ expressing HEK293 cells compared to controls, as well as most mtDNA-encoded proteins and many nuclear-encoded CI and CIV subunits and assembly factors [65]. These data demonstrate close spatial proximity of the mitochondrial import machinery, mitoribosomes, and insertase machinery to early respiratory CI intermediates as being important for the coordination of assembly of both the mtDNA- and nuclear-encoded OXPHOS subunits.

### 3.4. Plasticity of Mitochondrial Translation in OXPHOS Biogenesis

The coordination of mitochondrial translation with availability of nuclear-encoded factors is not well understood for most of the respiratory complexes; however, further study of the MITRAC complex demonstrated that, at least in the case of COX1 of CIV, availability of nuclear-encoded factors can regulate the rate of mitochondrial translation in a phenomenon termed translational plasticity [75]. C12orf62 was shown to specifically bind translating ribosomes that were translating *MT-CO1* mRNA [75] and was the strongest interaction of an MITRAC component with ribosomes, compared to weaker ribosome interaction of MITRAC12 or MITRAC15 [72] and no interaction with MITRAC7 [74]. Depletion of COX4, which is the first nuclear-encoded subunit to be added in the assembly of CIV, caused a stalling of *MT-CO1* translation and a build-up of nascent chains still attached to ribosomes and stabilized by C12orf62 [75]. This ribosome-arrested pool of *MT-CO1* can continue for full translation once the nascent COX1 protein is recruited into subsequent assembly steps when COX4 binds (Figure 2B) [75].

This translational plasticity process of regulating mitochondrial protein synthesis in humans is markedly different to that of yeast. In fact, several of the homologs of the MITRAC in yeast (e.g., Coa3 and Cox14) have the opposite effect on Cox1 translation, as they were shown to be negative regulators of Cox1 translation [76], whereas the human homologs (MITRAC12 and C12orf62) are required for COX1 translation [72]. MITRAC12 and C12orf62 are among the MITRAC components that were shown to interact with mitoribosomes [72,75], but yeast Coa3 and Cox14 do not [75], providing further evidence for the divergent functional evolution of these apparent homologs. This disparity is because humans and yeast have very different regulation of mitochondrial translation. In yeast, each mitochondrial transcript has specific translational activators that bind to the 5′ UTR of a particular messenger and promote translation. Using Cox1 as an example, one of the key the translational activators is Mss51; however, accumulation of Cox1 assembly intermediates, including the assembly factors Coa3 and Cox14, causes these assembly factors to inactivate Mss51 and stall translation of the *MT-CO1* mRNA (for a review, see Reference [77]). Human mitochondrial mRNAs do not contain 5′ UTRs and, therefore, do not have the same feedback regulation of mRNA translational activation and inhibition as yeast. The MITRAC complex and demonstration of translational plasticity is the only currently known example of specific mitochondrial translation regulation in humans [75] and is specific to the *MT-CO1* gene product, but far more is known about translation regulation in yeast for each of the mtDNA encoded subunits [78] including a recently described feedback loop involving several activators of cytochrome *b* [79].

### 3.5. Mammalian Translational Activator TACO1

MITRAC regulates translation at the insertion and assembly stage of nascent proteins, but the mechanism for stimulation of mammalian mitochondrial mRNAs translation is less known. As mentioned, mammalian mt-mRNA transcripts lack the 5′ UTRs that are present in yeast and are recognized by translational activators to regulate mitochondrial protein synthesis. However, a gene encoding the translational activator of cytochrome *c* oxidase 1, TACO1, was identified as a mitochondrial matrix protein and the first specific mammalian mitochondrial translational activator [80]. A single-base-pair insertion mutation traced back to TACO1 was found to associate with late-onset Leigh syndrome patients with isolated COX deficiency [80,81]. Mutant mouse models with homozygous mutations in TACO1 exhibited similar biochemical defects with reductions particularly in COX1 synthesis, assembly, and activity. Loss of TACO1 protein was linked to defective translation of *MT-CO1* because the steady-state levels of *MT-CO1* mRNAs were found to be unaffected in a mutant TACO1 mouse [82]. Using RNA electrophoretic mobility shift assays (EMSA), recombinant mouse TACO1 protein was found to bind *MT-CO1* mRNA at adenine–guanine-rich sites concentrated at the 5′ end. This provided strong evidence of the role of TACO1 specifically interacting with *MT-CO1* mRNA to promote *MT-CO1* translation (Figure 2C). Further experiments supported the role of TACO1 as a translational activator by revealing the association of TACO1 with the small subunits or monosomes of the mitochondrial ribosome, likely to facilitate initiation of *MT-CO1* mRNA translation. As COX1 is the first subunit to be incorporated in the early stages of CIV assembly, it is, hence, evident that TACO1 is important in the activation of COX1 synthesis to maintain normal levels of fully assembled CIV within the ETC [82].

## 4. Surveillance Mechanisms Regulating Mitochondrial Protein Import

Strict co-regulation of cytosolic and mitochondrial protein synthesis is necessary to protect cells from the accumulation of assembly intermediates in the mitochondrial milieu. Recent studies showed that cells have developed a number of protective quality control systems to regulate mitochondrial protein import, including constitutive and stress-induced mechanisms to remove arrested mitochondrial precursor proteins. The mitochondrial proteolytic system regulates a number of quality control activities, including degradation of misfolded/damaged polypeptides and processing of newly imported nuclear-encoded proteins by mitochondrial processing peptidases (MPP) - alongside its regulatory functions in OXPHOS complex assembly, phospholipid metabolism, and apoptosis. Mitochondrial quality control pathways are briefly discussed below, but with a primary focus on import stress responses. For more detailed reviews on mitochondrial quality control processes see References [83,84,85].

### 4.1. Mitochondrial Quality Control Pathways in Humans

An essential step for a large number of newly imported nuclear gene products is the removal of mitochondrial targeting presequence by MPPs. This leads to a release of import intermediates that require further processing by the mitochondrial intermediate peptidase (MIP) (Oct1 in yeast) to prevent destabilization of mature proteins. The subsequent clearance of presequence peptides is coordinated by the matrix pitrilysin metallopeptidase 1 (PITRM1), and studies demonstrated that PITRM1 impairment triggers accumulation of presequence peptides [86]. The yeast homolog of PITRM1, Cym1, together with the homolog of the human insulin-degrading enzyme (IDE), Ste23, were shown to have a cooperative role in peptide clearance [87]. Indeed, it was proposed that defects in this step have a negative impact on MPPs, ultimately leading to accumulation of newly imported unprocessed polypeptides [86,87].

The recent discovery of the stress-induced metalloendopeptidase OMA1 proteolysis pathway allowed further insight into human mitochondrial quality control mechanisms [88]. The AAA (ATPases associated with diverse cellular activities) protease complexes are one of many proteolytic enzymes employed to maintain proteostasis within the mitochondria, regardless of whether the protein was imported from the cytosol or translated by localized mitoribosomes in the matrix. The AAA protease complexes in humans are transmembrane protein complexes in the IMM which are divided into two types, the matrix-facing m-AAA protease and the i-AAA protease which projects its catalytic site toward the intermembrane space (IMS). The m-AAA proteases are either homohexamers of the ATPase family gene 3-like protein 2 (AFG3L2) or heterohexamers of AFG3L2 and the protein paraplegin (encoded by *SPG7*), whereas the i-AAA protease is formed by ATP-dependent zinc metalloprotease YME1L. Substrates for the AAA proteases are proteins with unstructured terminus of about 8–10 amino acids which are then targeted to complete degradation or chaperoned for refolding to be assembled in the OXPHOS complexes [89]. Disruption of the AAA protease-mediated mitochondrial matrix quality control pathway is associated with mitochondrial dysfunction. This is caused by the accumulation of nascent protein chains that are not actively removed due to defective AAA proteolytic activity [88,90]. Ehses and colleagues proposed that the disruption to the mitochondrial AAA proteolytic quality control pathway induces the activation of OMA1 which cleaves the IMM-anchored GTPase, optic atrophy 1 (OPA1) protein, to be released as a soluble form in the mitochondrial matrix [91]. This mitochondrial stress response causes alteration in the mitochondrial membrane morphology to downregulate protein synthesis through mitoribosome and mRNA decay without affecting mtDNA integrity or copy number. OMA1 activation is a specific proteolytic response to defective mitochondrial protein synthesis to transduce a negative feedback cascade on the organelle form and function in the absence of other mitochondrial matrix quality control mechanisms [88,92].

Recent studies in human cell lines focusing on OXPHOS complex assembly and turnover rates of mitochondrial proteins using SILAC pulse-labeling provided new insights not only into the dynamics of the mitoribosome [46] but also OXPHOS (CI, CIV and CV) assembly [93]. As described earlier, many MRPs are produced and imported into mitochondria in excess [46,93]. Although this may suggest that over-synthesis of MRPs is important in order to ensure effective mitoribosome biogenesis by providing a sufficient supply of its subunits, it may also be inefficient and disadvantageous, as accumulation of unused MRPs requires the activation of selective protein degradation systems and could also lead to mitochondrial damage caused by proteotoxicity [46]. Interestingly, the latter study found differences in the rates of newly synthesized and imported OXPHOS subunits [93]. In particular, the synthesis rate of CI subunits facing the matrix was dramatically increased, and these proteins underwent degradation within 3 h. The identified matrix-exposed subunits corresponded mainly to the N- and Q-modules of CI, and the excess may be represented by the accumulation of partially assembled subcomplexes of CI. Whilst the cytochrome *c* oxidase (COX) subunit NDUFA4 was the only over-synthesized subunit of CIV, the assembly of CV was very efficient with low excess subunit synthesis [93]. Clearly, these data suggest that there are mechanisms in place that control the turnover rate of the excess MRPs and specific OXPHOS subunits in order to prevent proteotoxicity within the organelle through mitochondrial quality control systems; however, these pathways remain to be fully elucidated.

Thus, in addition to the carefully balanced regulation of cytosolic and mitochondrial translation programs, it is evident that protective quality control mechanisms are necessary (i) to prevent the accumulation of mitochondrial precursor proteins, and (ii) to regulate the rate of degradation of excess protein import into the mitochondria in order to maintain organelle homeostasis.

### 4.2. Mitochondrial Import Stress Response Pathways in Yeast

The entry site of almost all nuclear-encoded mitochondrial proteins is coordinated via the TOM complex before being passed on to the TIM23 complex. It was demonstrated that disturbance of mitochondrial protein import leads to the activation of stress response signaling pathways, which is discussed in more detail below. Many stress response pathways were studied in yeast, such as the UPRam (an unfolded protein response activated by mistargeting of proteins leading to increased proteasomal activity) [94] or the mitochondrial precursor overaccumulation stress pathway (mPOS) [95]. Using a ADP/ATP carrier 2 (AAC2) mutant yeast model of a previously reported human mitochondrial disease gene *SLC25A4* [96], Wang and Chen demonstrated that the mPOS pathway is triggered not only by abnormal accumulation of mitochondrial precursors in the cytosol, but also by severe mitochondrial damage that directly contributes to the proteostatic stress and degenerative cell death [95].

Recently, a surveillance mechanism termed mitoCPR (mitochondrial compromised protein import response) was identified at the mitochondrial surface to protect mitochondria from an overload of precursor proteins [97]. During mitochondrial import stress, the activation of the mitoCPR pathway results in a pleiotropic drug resistance 3 (Pdr3)-mediated transcriptional response and expression of *CIS1* (cistrinin resistance protein), a cytosolic gene product that associates with the peripheral OMM. Cis1 binds the Tom70 mitochondrial translocase receptor and decreases the levels of precursor proteins in conjunction with the AAA+ ATPase Msp1 that promotes proteasomal degradation of stalled proteins [97]. However, only recently a mitochondrial quality control system that continuously monitors and prevents the accumulation of precursor proteins at the mitochondrial entry gate under non-stress conditions was identified [98]. The mitochondrial protein translocation-associated degradation pathway named mitoTAD ensures constitutive quality control via an adaptor ubiquitin regulatory X (UBX) domain-containing protein 2 (Ubx2), which functions in both the mitochondrial and endoplasmic reticulum (ER)-associated degradation pathways. The mitochondrial pool of Ubx2 binds the TOM complex via Tom70 that results in the recruitment of the AAA+ ATPase Cdc48 to facilitate constitutive clearance of TOM-arrested mitochondrial precursor proteins [98].

The OMM-associated homeostasis systems seem to also converge with the cytosolic ribosome-associated quality control (RQC) pathway in order to prevent accumulation of stalled nuclear-encoded mitochondrial translation products under non-stress conditions [99]. A mitochondrial RQC quality control system (mitoRQC) protects against such toxicity by the synergistic action of the cytosolic RQC factor Vms1 and the Ltn1 E3 ubiquitin ligase [99]. Vms1 sequesters aberrant mitochondrial proteins to be degraded via *in organello* protein quality control systems. Indeed, deletion of both *VMS1* and *LTN1* in yeast leads to an increase of insoluble proteins including subunits of the OXPHOS system, MRPs, and various chaperones, suggesting that the mitochondrial protein synthesis is impaired, which is reflected by the subsequent defect in respiratory chain Complexes III, IV, and V [99].

Interestingly, the yeast Vms1, Ubx2, and Msp1 proteins were identified as interacting partners [98], implying that cross-talk between distinct mitochondrial import stress response pathways exist. It remains to be determined whether such protein surveillance mechanisms take place in humans; however, the conservation of many of the components of the protein quality control pathways is making it an attractive area to study.

## 5. The OXPHOS System and Formation of Its Early Assembly Intermediates

With intricate orchestration of cytosolic and mitochondrial gene expression for protein synthesis and import, the assembly of mitochondrial OXPHOS complexes also requires a wide range of assembly factors to optimize the process. The functional and structural organization of mitochondrial respiratory chain complexes into bigger entities was widely debated for many years. Historically, individual members of the respiratory chain complexes were thought to be in a rigid arrangement, which involves tight coupling of the protein complexes to enhance mitochondrial respiration through electron transport [100,101]. Following the purification of enzymatically active complexes, the “solid model” was challenged by a “fluid model” which suggested that random collision of freely diffusing mitochondrial inner membrane proteins and electron carriers can facilitate electron transport based on protein concentration and ionic strength [102]. However, the most recent “plasticity model” is now widely accepted where ETC complexes can exist both individually and also assembled into large units commonly referred to as supercomplexes or respirasomes, to increase proximity and, therefore, improve efficacy of electron transport during OXPHOS [103,104]. Interestingly, a recent study revealed that CIII_2_ dimers in the supercomplexes are crucial for the maturation of CI and CIV, challenging the idea that individual complexes could be assembled individually before forming supercomplexes. It was proposed that CIII_2_ in supercomplexes facilitates the assembly of CI functionally through the redox balance of CoQ pool and by structurally binding to the pre-CI intermediates. CIV subunits are sequestered in CIII subassemblies until supercomplexes are formed to allow complete assembly of CIV [105]. Clearly, the kinetic behavior of the respiratory chain complexes is not random, and it requires highly intricate coordination to optimally modulate and adapt to various cellular requirements. For a detailed review on the formation and function of supercomplexes, see Reference [106]. Additionally, the mitochondrial cristae structure was found to be essential for the assembly and functionality of the OXPHOS system [107]. Among a large family of proteins that shape the cristae, OPA1 and the mitochondrial contact site and cristae-organizing system (MICOS) complex are the master regulators of cristae dynamics. Genetic manipulation of both OPA1 and MICOS1 was shown to perturb cristae structure, subsequently impacting the formation of OXPHOS complexes and supercomplexes. For expert reviews on the relationship between cristae dynamics and mitochondrial OXPHOS bioenergetics, please refer to References [108,109].

With the advancement in high resolution cryo-EM, computational protein modeling, and complexome profiling analysis in recent years, the human OXPHOS protein complexes and their assembly were analyzed and described in greater detail with reference to the other model organisms, which are discussed next. OXPHOS complex assembly was comprehensively reviewed in Reference [110].

### 5.1. Complex I (NADH–Ubiquinone Oxidoreductase) Assembly

CI has 44 different subunits of which seven are encoded by the mtDNA (*MT-ND1-6* and *MT-ND4L*), while the remaining subunits originate from nuclear DNA [111,112]. CI is an L-shaped complex that has two distinct domains: the hydrophilic head projecting into the matrix and the hydrophobic part that lies within the IMM [113]. As the first entry point of electrons into the ETC, CI catalyzes the transfer of two electrons through NADH oxidation to the electron carrier, ubiquinone [114,115]. This process is coupled with the translocation of four protons into the IMS through piston-like motion exerted by conformational changes upon quinone binding and potentially also by networking between charged residues to pump out the fourth proton [116].

CI assembly is a sequential process that requires pre-assemblies of three main intermediate modules, namely, the N-module, Q-module, and P-module. The N-module makes up the head of the hydrophilic arm containing the NADH-binding site and a flavomononucleotide (FMN) group which oxidizes NADH to release two electrons [117]. A known assembly factor of the N-module is NDUFAF2, which is associated with final stage assembly but was also proposed as a chaperone for accurate protein folding to ensure structural stability of the N-module [118,119]. The Q-module residing in the hydrophilic arm interfacing hydrophobic membrane components of CI contains eight Fe–S clusters where electrons are passed down before being transferred to ubiquinone [117]. NDUFAF3 and NDUFAF4, in addition to NDUFAF5 and NDUFAF7, both of which are *S*-adenosylmethionine (SAM)-dependent enzymes that post-translationally modify CI core subunits, are involved in the biogenesis of the Q-module [120,121,122]. On the other hand, NUPBL is responsible for Fe–S cluster incorporation following stabilization of the Q-module [122]. The P-module, which houses all seven mtDNA-encoded CI subunits, corresponds to the hydrophobic membrane arm. Its assembly is further grouped into proximal submodules (P_P_) and distal submodules (P_D_). The P_P_ submodule is further divided into two intermediates which are P_P-a_ (also known as the ND1-module) and P_P-b_ (ND2-module) [117,122]. The Translocase of inner mitochondrial membrane domain containing 1 (TIMMDC1) assembly factor participates in P_P-a_ assembly by allowing association with the other modules through subunit binding, in particular with mitochondrial complex I assembly (MCIA) complex [123,124,125]. P_P-b_ submodule assembly relies on the MCIA complex constituted by NDUFAF1, ECSIT (Evolutionarily conserved signaling intermediate in Toll pathways), ACAD9, and TMEM126B (Transmembrane protein 126B) as the cornerstone of CI assembly [117]. TMEM186 and MITRAC15 were found to associate with the MCIA complex stabilized by ACAD9 centrally, to facilitate CI assembly [126]. P_D_ submodules of the membrane arm can also be divided into two intermediates, P_D-a_ (ND4-module) and P_D-b_ (ND5-module). The assembly factors that take part in P_D_ submodule assembly are FOXRED1 (FAD Dependent Oxidoreductase Domain Containing 1), DMAC1 (Distal membrane-arm assembly complex protein 1), and ATP5SL (Distal membrane-arm assembly complex protein 2), which interact with each other for late-stage assembly of the P-module and possibly its membrane insertion by interacting with OXA1L [122,127,128,129]. The elucidated order for CI modular assembly begins with P_P-b_ associating with P_D-a_, followed by the joining of P_D-b_ or Q/P_P-a_ intermediates to form Q/P intermediates, and finally addition of the N-module for CI maturation [122,123]. It is important to note that there are a number of assembly factors and accessory units that do not associate with a particular module but rather interact at the interface of two modules to aid CI assembly intermediate formation, while some may be released during the assembly process to allow further maturation of the complex [129].

### 5.2. Complex II (Succinate–Ubiquinone Oxidoreductase) Assembly

CII is the only OXPHOS component which does not contribute to proton pumping across the mitochondrial membrane. All four subunits of CII are encoded by the nuclear genome, consisting of hydrophobic membrane-anchoring proteins and the hydrophilic catalytic domain. CII has dual roles in mitochondria: (i) in the TCA cycle, it metabolizes succinate to fumarate, and (ii) in the ETC, it facilitates electron transfer from flavin adenine nucleotide (FAD) reduction to electron carrier ubiquinol [130,131].

The hydrophilic domain of CII is a heterodimer of SDHA (Succinate dehydrogenase complex flavoprotein subunit A) and SDHB subunits which are assembled in parallel. SDHAF2 covalently attaches FAD which would be reduced in the event of succinate oxidation in the SDHA subunit [132,133]. FAD reduction releases two electrons which are transferred through three Fe–S clusters incorporated by SDHAF1 containing an LYR (Leu–Tyr–Arg) motif that interacts with Fe–S clusters on its N-terminus while binding to SDHB at its C-terminus [134,135,136]. SDHAF3 plays a protective role for the Fe–S clusters in SDHB against reactive oxygen species (ROS) [136,137]. Flavinated SDHA forms a heterodimer with SDHB in the presence of SDHAF4 which also protect the subunit from oxidative stress [137,138,139]. The hydrophobic membrane anchor of CII is made up of SDHC and SDHD contributing to the quinone binding site, providing reduction of ubiquinone to ubiquinol, the mobile electron carrier that links to CIII. Although the assembly pathway for the membrane anchor domain is yet to be revealed, it was shown that the assembly of the hydrophobic subunits are influenced by SDHA and SDHB biogenesis [139,140]. SDHAF1 and SDHAF3 were shown to play a role in the joining of SDHB subunit to the membrane anchor in IMM [136]. Interestingly, an alternative assembly pathway of CII was identified that describes the slower migrating CII (~100 kDa), referred to as CII_low_. The CII_low_ intermediate is lacking SDHB and SDHC. However, in response to bioenergetic stress, particularly when mtDNA is depleted, CII_low_ associates with SDHA and SDHAF2 and SDHAF4 assembly factors. This CII_low_ formation enables adaptations to reduce energy-demanding cellular processes such as *de novo* pyrimidine synthesis to halt the cell cycle at the S phase and reduce anabolic processes even in the TCA cycle [141].

### 5.3. Complex III (Ubiquinol–Cytochrome c Oxidoreductase) Assembly

CIII or the cytochrome *bc_1_* complex has a total of 11 subunits. Oxidation of ubiquinol to ubiquinone takes place at the only mtDNA-encoded subunit, cytochrome *b*¸ generating two electrons that are channeled to two distinct routes. One to the Rieske Fe–S protein UQCRSF1, followed by electron transfer to cytochrome *c* through the cytochrome *c_1_* subunit with a heme *c_1_* metal center, resulting in pumping of two protons across IMM. Another electron passes through the low-potential path to catalyze recycling of ubiquinol through reduction of ubiquinone or semiquinone. However, the reduction of ubiquinol has to be repeated for full turnover of ubiquinol, thus allowing another electron to be transferred to cytochrome *c_1_*, and a total of four protons to be translocated into the IMS [142].

The assembly of a functional CIII_2_ dimer is initiated by the insertion of its mtDNA-encoded catalytic subunit cytochrome *b* (CYTB) into the IMM by UQCC1 and UQCC2 bound at the exit tunnel of mitoribosomes [143,144]. UQCC3 is in charge of the subsequent insertion of heme *b* groups at the *b_L_* and *b_H_* heme-binding site in cytochrome *b* [145,146]. Addition of *c*-type heme in cytochrome *c_1_* is facilitated by the cytochrome *c* heme lyase (CCHL) protein in the IMM [147,148]. Following incorporation of respective heme groups, cytochrome *c_1_* associates with cytochrome *b* in the presence of UQCRH for dimerization, forming a pre-CIII_2_ dimer [149,150]. Finally, CIII_2_ assembly is completed with the addition of the Rieske Fe–S protein, UQCRFS1, and two other accessory subunits [151,152,153,154]. In yeast, Mzm1 stabilizes and inserts the Fe–S cluster into the Rip1 subunit (yeast homolog of UQCRFS1) [155,156], whilst, in humans, LYRM7 aids the Fe–S incorporation into UQCRFS1 [157]. A unique feature of human CIII_2_ is that the N-terminal peptide of UQCRFS1 is cleaved to release mature protein and the mitochondrial targeting sequence remains in the complex as an accessory subunit called SU9 [158]. The processing of UQCRFS1 is hypothesized to take place after the incorporation of the protein into the pre-CIII_2_ complex by UQCRC1 and UQCRC2, which has matrix processing peptidase (MPP) activity [150]. TTC19 is responsible for the removal of N-terminal peptides generated from UQCRFS1 processing to allow the maturation of the subunit and completing the assembly of CIII_2_ [150,159].

### 5.4. Complex IV (Cytochrome c Oxidase) Assembly

CIV (cytochrome *c* oxidase) comprises 14 subunits, with all three catalytic subunits (COX1, COX2, and COX3) encoded by mtDNA. CIV represents the final stage of electron transfer across the ETC by cytochrome *c* to oxygen as the final electron acceptor in OXPHOS [160]. It possesses redox active cofactors (Cu_A_ and binuclear Cu_B_ centers) and heme prosthetic groups to couple electron transport with proton pumping across the complex. The Cu_A_ center in the COX2 subunit accepts electrons carried by cytochrome *c* from CIII. The heme *a* group in the membrane-spanning COX1 subunit is responsible for the delivery of electrons from Cu_A_ to the oxygen-reducing heme *a_3_*-Cu_B_ center. The electron transfer leads to the pumping of a total of four protons into the IMS [161,162], which was suggested to be coupled with the reduction of COX subunits even at the early stages of electron transfer to the Cu_A_ center [163]. COX3 is proposed to have a structural role in stabilizing the other two core subunits and proton pumping [164].

The assembly process of CIV is initiated by the nuclear-encoded subunits, COX4 and COX5A, in association with the HIG1 domain family member 1A (HIGD1A, human homolog of yeast Rcf1) [165,166,167]. However, the membrane insertion of COX1 chaperoned by the earlier mentioned MITRAC complex (C12orf62 and MITRAC12), in which the COX assembly mitochondrial protein homolog encoded by CMC1 joins as a stabilizing factor, is still recognized as the basis for CIV assembly [75,168,169,170]. The heme *a* group in the membrane-spanning COX1 subunit is synthesized by COX10, and COX15 oligomers synthesize heme *a* for COX1 and associate with Pet117, which facilitates heme *a* biosynthesis and delivery [171,172,173,174]. The COX assembly factor SURF1 (Surfeit locus protein 1) was first implicated in heme *a* delivery, but later studies suggested its involvement in heme assembly or stabilization of heme binding site instead [175,176]. SURF1-deficient mice activate mitochondrial stress responses, such as the mtUPR (mitochondrial unfolded stress response) [177], while pathogenic variants in *SURF1* are linked to COX-deficient Leigh syndrome in human resulting from defective CIV assembly and the accumulation of subcomplexes [178,179]. Copper ions for CIV redox cofactor centers are transported by metallochaperone COX17 to either COX11 for the metalation of COX1 Cu_B_ center or SCO1/2 (synthesis of cytochrome *c* oxidase 1/2) through loop recognition, where they cause disulfide reduction for copper binding in the Cu_A_ center of COX2 [180,181,182,183,184]. The binuclear Cu_A_ center in the COX2 soluble globular domain needs to be translocated toward the IMS by COX18 [170,185]. It was demonstrated that COA6 has thiol-reductase activity, catalyzing cysteine reduction in SCO1 and SCO2 to facilitate copper insertion in COX2 subunit [186]. The metalation of the Cu_A_ center in COX2 is also aided by COX16 that promotes incorporation of the subunit into the MITRAC complex [187,188,189,190]. COX1 intermediates are stabilized by MITRAC15 (COA1) and MITRAC7 prior to association with COX2 [74,191]. Late assembly of CIV is largely uncharacterized to date, but a vertebrate-specific protein MR-1S was elucidated to interact with Pet117 and Pet100 for the assembly of the remaining nuclear-encoded COX subunits and COX3 [165]. NDUFA4 which was initially thought to be part of CI, is now shown to be the last subunit to be incorporated into the CIV holoenzyme [192]. A number of COX subunits have different isoforms depending on tissue type, likely related to the energy demand [176]. As the assembly kinetics are mostly understood from conserved proteins in yeast and other model organisms, it is believed that there are still unknown assembly factors that take part in different stages of CIV assembly.

### 5.5. Complex V (F_1_F_O_ ATP Synthase) Assembly

CV is responsible for the production of ATP, utilizing the proton-motive force generated through electron transfer. The protein complex has 18 subunits (including the regulatory protein IF1 (Inhibitory factor 1)), in which only ATP6 and ATP8 of the F_O_ domain are encoded by the mitochondrial genome. The F_1_F_O_ ATP synthase has two functional domains: the hydrophilic matrix-facing F_1_ domain for ATP generation and the membrane-facing F_O_ domain for proton translocation. The structure of CV is supported by a central axis and a peripheral stalk preventing unnecessary rotation to maximize efficacy of ATP synthesis. The proton translocation leads to rotary motion of the c-ring in the F_O_ domain that is connected to the F_1_ catalytic subunit by the central stalk. Coupling of the rotation confers energy for the enzyme to drive ADP condensation with Pi to ultimately synthesize ATP required as an energy source for various cellular processes [193,194].

The F_1_-c rotor is an important assembly intermediate that was consistently identified in different genetic knockout experiments to characterize CV assembly, signifying the basis of ATP synthase protein assembly [195,196,197]. The complete assembly of F_1_ domain itself requires two subassemblies, which are the α_3_β_3_ hexamers and the γδε complex that make up the central stalk [198]. Whilst most of the assembly factors involved in ATP synthase assembly are unknown, ATP5AF1 and ATP5AF2 are associated with the formation of α_3_β_3_ hexamers [199,200]_._ The oligomerization of c subunits to form the rotor c-ring is possibly facilitated by TMEM70 to assemble with the F_1_ domain [201,202]. Interestingly, TMEM70 was also linked to CI assembly, where the protein comigrated with the P_D_ module and knockout cells accumulated Q/P_p_ intermediates, suggesting a role in the stabilization of CI P-module during assembly [203]. The peripheral stalk consisting of b, d, F6, and OSCP subunits binds to the F_1_-c complex for stabilization, followed by association with subunits e, f, g [195,196,197]. The assembly intermediates to this point are bound by IF_1_, which is an inhibitor of protein hydrolase activity before full CV is formed. IF_1_ dissociates when mtDNA-encoded subunits, a/ATP6 and A6L/ATP8, bind to the complex in the final steps of CV assembly [197,204]. The 6.8PL accessory factor was identified as a supernumerary subunit that stabilizes ATP6 and ATP8 incorporation in CV and the activator of ATP synthase [197,205]. Furthermore, *ATP5MD* encoding the DAPIT (Diabetes-associated protein in insulin-sensitive tissue) CV subunit also plays an important role in the dimerization of ATP synthases within the IMM, in order to form rows of V-shaped dimers causing more than 70° bending of the membrane, which is important for proper cristae formation to enhance the efficacy of mitochondrial ATP synthesis [197,206,207].

## 6. Final Stages of OXPHOS Assembly

The intricacy of mitochondrial OXPHOS complex biogenesis is evident by the involvement of many tightly orchestrated steps and factors required to form a fully functional respiratory chain machinery. To aid the biogenesis of these complexes, cells evolved complex-specific assembly factors, many of which are conserved from yeast to humans. The optimal assembly of OXPHOS complexes requires both early- and late-stage ancillary proteins. The early-stage assembly factors, as described above, have critical roles in the structural assembly of individual subunits and subcomplexes. However, the late-stage accessory factors also known as the LYRM (leucine–tyrosine–arginine motif) proteins regulate the incorporation and/or activation of final subunits and/or co-factors such as the Fe–S clusters to provide optimal assembly for CI, CII, CIII, and CV.

### 6.1. The LYRM Protein Family

LYRM proteins are small (~10–22 kDa) basic polypeptides that belong to the Complex1_LYR-like protein superfamily. The main characteristic feature of LYRM proteins is the conserved leucine–tyrosine–arginine (LYR) sequence at the N-terminus and the conserved phenylalanine and arginine residues located downstream of the LYR triplet. The human mitochondrial proteome contains at least 12 LYRM proteins with functions in OXPHOS complex assembly (LYRM2, LYRM3, LYRM6, LYRM7, LYRM8, ACN9 (Acetate non-utilizing protein 9), and FMC1 (Formation of mitochondrial complex V assembly factor 1)), Fe–S cluster biosynthesis (LYRM4), mitoribosome assembly (L0R8F8), regulation of the electron transferring flavoprotein ETF (LYRM5), and two uncharacterized LYRM proteins, LYRM1 and LYRM9 (Figure 3). Large-scale protein interactome mapping and high-resolution cryo-EM structural studies demonstrated that members of the LYRM family interact with an acyl carrier protein (ACP) that, in humans, also serves as a structural subunit of CI [47,112,208,209,210].

### 6.2. Interactions between Mitochondrial Acyl Carrier Proteins and LYRM Proteins

In eukaryotes, mitochondrial ACPs (mtACPs) have essential functions in the mitochondrial type II fatty acid synthesis pathway (mtFAS-II), which highly resembles the fatty acid synthesis pathway found in prokaryotes [211]. The mtACP acts as a soluble scaffold protein for the synthesis of fatty acids from acetyl-CoA. By utilizing its 4′-phosphopantetheine (4′-PP) prosthetic group, mtACP can covalently attach to a conserved serine residue and, thus, provide an attachment point for the growing fatty acyl chain through thioester linkage (for a review, see Reference [212]). Originally, it was suggested that the main function of the mitochondrial mtFAS-II pathway was to synthesize lipoic acid from the eight-carbon octanoyl-ACP unit, a reaction catalyzed by the lipoic acid synthase. Endogenous lipoic acid is an essential cofactor in mitochondrial metabolism, regulating the enzymatic activity of 2-ketoacid dehydrogenase complexes, including branched-chain ketoacid dehydrogenase, pyruvate dehydrogenase, and the key regulatory enzyme of the TCA cycle, α-ketoglutarate dehydrogenase [213,214]. In line with this, mice deficient in lipoic acid synthase undergo early embryonic lethality [215], and pathogenic variants identified in the human lipoic acid synthase (*LIAS*) are the cause of severe metabolic crisis accompanied by hyperglycinemia, lactic acidosis, and seizures [216,217]. However, beyond its function in lipoic acid synthesis, structural analysis of human mtFAS-II pathways and in vitro kinetic studies suggest that the mtFAS-II pathway is capable of generating fatty acids longer than eight carbons [218,219].

More recently, a study focusing on mtACPs from *Yarrowia lipolytica*, ACPM, demonstrated that, although only a minor fraction of ACPMs were present in their free form, the majority of ACPMs were bound to CI and small amounts were found in the LYRM4/NFS1 complex that facilitates Fe–S cluster biogenesis. Interestingly, only the CI-bound fraction of ACPMs was post-translationally modified by the attachment of medium and long acyl groups (C10–C16). Therefore, the authors hypothesize that acyl groups longer than 10 carbons can serve as a docking station for ACPMs and LYRMs, and such form of acyl modification is required for correct CI assembly [220]. In addition, structural studies of mtACP/LYRM complexes provide further evidence suggesting that mtACP inserts an extended 4′-PP-acylated group into the hydrophobic pocket of the LYRM protein [47,112,221], thus establishing a unique interaction between mtACP and LYRM. This fraction of mtACP is likely to function in the assembly of respiratory chain complexes rather than lipoic acid synthesis.

Subsequent studies in *Sacharomyces cerevisiae* demonstrated that LYRM proteins require an acetyl-CoA dependent allosteric activator of the LYRM network in order to stimulate OXPHOS biogenesis [222]. It was proposed that the acylation of mtACP is essential for LYRM protein interaction via its Leu–Tyr–Arg motif, as these proteins possess a higher affinity for the acylated form of mtACP than the deacylated form, suggesting that acylation of mtACP could act as a regulator of LYRM proteins to activate the OXPHOS assembly [222]. One remaining question is why this post-translational modification, in the form of mtACP-acylation, activates OXPHOS assembly and whether this is applicable to higher eukaryotes.

### 6.3. The Role of mtACP and LYRM Proteins in OXPHOS Assembly

#### 6.3.1. Complex I

The mtACP protein, also known as NDUFAB1 in humans, was originally discovered as an accessory subunit of CI in bovine heart mitochondria [223]. Since then, it is now clear that NDUFAB1 not only participates in CI assembly, but it also coordinates a catalog of mitochondrial functions. More recently, NDUFAB1 interacting partners were captured by affinity enrichment mass spectrometry (AE-MS) proteomics studies confirming the NDUFAB1–LYRM interaction network [209]. Although many of these interactions are still not well characterized, it is clear that NDUFAB1 is a central regulator of LYRM proteins.

Stroud et al. explored the importance of 31 CI accessory subunits using CRISPR/Cas9 and TALEN gene editing strategies to generate human knockouts of each subunit [129]. Interestingly, whilst 25 subunits were found to be required for the assembly of a functional CI, only NDUFAB1 was indispensable for cell viability. Re-expression of the human NDUFAB1 or the yeast mtACP in the NDUFAB1 knockout background resulted in cell survival in glucose-rich media. However, when forcing cells to rely on mitochondrial respiration, rather than glycolysis, by culturing cells in galactose-rich media, only complementation with human NDUFAB1 supported cell growth. Therefore, these data suggest that, although complementation with the yeast mtACP is sufficient to support cell growth upon loss of NDUFAB1, mtACP is not able to rescue the essential function of NDUFAB1 in CI assembly and, therefore, it functions independently of CI during cell survival [129].

There is growing evidence that loss of NDUFAB1 in mammals not only results in CI deficiency, but also destabilization of CII, CIII, and supercomplex formation [224]. To date, no disease-causing mutations were identified in NDUFAB1, and a full-body knockout mouse model of NDUFAB1 is embryonic lethal. However, a cardiac-specific NDUFAB1 knockout mouse model was shown to cause dilated cardiomyopathy resulting in heart failure by ~12 weeks of age [224]. The importance of NDUFAB1 in cardiac function was demonstrated by the significant decrease in the amounts of Complexes I, II, and III, as well as supercomplexes, in the absence of NDUFAB1. As expected, the steady-state levels of CI subunits were decreased; however, only the levels of the Fe–S-containing subunits of CII (SDHB) and CIII (UQCRFS1) were markedly decreased, whilst the levels of other CII and III subunits remained unchanged. It is worth mentioning that no CIV defect was observed in the cardiac-specific NDUFAB1 knockout mice [224]. These data provide an interesting hypothesis that NDUFAB1 specifically coordinates the assembly of Fe–S-containing complexes—not including CIV—by regulating the biogenesis of Fe–S clusters.

High-resolution Cryo-EM studies in mammals clearly defined the structure of CI and pinpointed the presence of the only accessory subunit, NDUFAB1, which is present in two distinct locations [112,225]. One copy of the NDUFAB1 subunit interacts with LYRM3 (NDUFB9) within the membrane arm of CI and another copy is in contact with LYRM6 (NDUFA6) within the peripheral arm of the complex. These interactions are critical to maintain CI assembly and stability. In particular, the presence of long acyl chains (C10–C16) attached to the 4-PP’ prosthetic group of the ACP (NDUFAB1) is essential for anchoring of mtACP (NDUFAB1) to CI via the LYRM–mtACP interaction [220,226].

Recent AE-MS quantitative proteomics profiling of the NDUFAB1 interaction network confirmed nine known LYRM interacting partners (LYRM2, LYRM3, LYRM4, LYRM5, LYRM7, SDHAF3, FMC1, L0R8F8, and LYRM9) [210]. Functional proteomics and blue native gel electrophoresis (BN-PAGE) analysis of the LYRM2 knockout cell line showed a specific CI defect that was associated with a decrease in the assembly of CI and CI-containing supercomplexes. Analysis of the topographical proteomic heat map of CI in the LYRM2 knockout cell line revealed a defect in the final stages of CI assembly that was marked by the absence of the N-module. Although the exact function of LYRM2 in CI assembly remains elusive, it could be hypothesized that a transient interaction of LYRM2 with CI is essential during the maturation of the N-module where Fe–S and flavin mononucleotide (FMN) cofactors reside [210].

#### 6.3.2. Complex II

Two mammalian LYRM proteins are associated with CII maturation, LYRM8 (SDHAF1) and ACN9 (SDHAF3). SDHB is the only structural subunit of the tetrameric CII that contains three Fe–S centers. LYRM8 aids the incorporation of the three Fe–S clusters into SDHB via an interaction with the co-chaperone HSC20 (Heat shock cognate protein 20) of the cytosolic iron sulfur cluster scaffold- heat shock 70-kDa protein 9 (cISCU–HSPA9) [227]. Although LYRM8 does not directly participate in the incorporation of the Fe–S clusters into the SDHB subunit, it serves as a guide that assists the delivery of these essential cofactors. Both LYRM8 and ACN9 were suggested to protect the Fe–S clusters against ROS damage during assembly. Indeed, loss of either of the CII LYRM factors in eukaryotes results in SDH deficiency and susceptibility of SDH to the harmful effects of endogenous oxidants [136].

#### 6.3.3. Complex III

The final assembly stages of CIII require the incorporation of two Fe–S clusters into the UQCRFS1 Rieske protein. Mzm1 was first identified in yeast as an LYRM protein involved in the incorporation of Rip1, the yeast homolog of UQCRFS1, into CIII_2_ dimers [155]. Yeast with a mutated LYR motif have a CIII assembly defect and decreased CIII activity, which is in line with the requirement of the Leu–Tyr–Arg sequence to mediate the LYRM–mtACP interaction [155,222]. Similarly, pathogenic mutations in the human LYRM7 (MZM1L) are associated with early-onset mitochondrial disease and severe isolated CIII deficiency [228]. Both human LYRM7 (MZM1L) and yeast Mzm1 were shown to bind the Rieske protein UQCRFS1 and Rip1, respectively, in order to stabilize it prior its incorporation into the late CIII_2_ dimer intermediate [155,157].

#### 6.3.4. Complex V

FMC1 is an atypical LYRM protein, belonging to the Complex 1_LYR_2 family. In yeast, Fmc1 functions as a CV assembly factor, stabilizing the yeast homolog of ATPAF2, Atp12. Direct interaction between the mtACP and FMC1 was shown in both yeast and humans [209,210,229]. However, yeast studies demonstrated that abrogating the mtACP–LYRM interaction by introducing a mutation in the LYR motif of *FMC1* does not have an impact on growth or CV assembly, suggesting that Fmc1 function may be ACP-independent [222].

### 6.4. The Role of mtACP and LYRM Proteins in Fe–S Cluster Biogenesis

The biosynthesis of Fe–S clusters is an essential and versatile process providing cofactors for numerous enzymes and cellular processes. In eukaryotes, Fe–S cluster biogenesis requires multiple assembly steps coordinated via the iron sulfur cluster (ISC) machinery. The initial phase of Fe–S cluster assembly requires the ISCU scaffold protein. The iron donor protein frataxin (FXN) provides the iron atom, and the sulfur is delivered by the desulfurase complex consisting of the cysteine desulfurase NFS1 and LYRM4 (ISD11). Following further maturation, the ISCU/HSC20/HSPA9 complex facilitates the transfer of Fe–S clusters to their targets (for a review, see Reference [230]). Yeast LYRM4 mutants (*ISD11*) with abolished LYR motif have partially impaired Nfs1 function and are prone to aggregation. In humans, pathogenic mutations in LYRM4 result in multiple OXPHOS defects affecting Fe–S-containing complexes I, II, and III [231]. The mtACP interaction with LYRM4 (ISD11) was described as an important factor in the Fe–S biogenesis, where, by exploiting the 4-PP’ acyl chain, mtACP stabilizes the NFS1–ISD11 complex, which is otherwise prone to degradation [232].

### 6.5. The Role of mtACP and LYRM Proteins in Mitoribosome Assembly

Recent reconstitution of the human mitochondrial ribosome structure in a native state of assembly revealed the presence of two new mitoribosomal assembly factors [47]. Cryo-EM analysis at ~3Å resolution of the late-stage mitoribosome intermediates uncovered an additional density within the mt-LSU located next to the mitochondrial ribosomal protein L14 (uL14m) subunit. Half of the additional density corresponded to a previously characterized protein MALSU1, a mitoribosomal assembly factor that facilitates the incorporation of uL14m into the mt-LSU, which is an essential step in the maturation of the mt-LSU [233,234]. Surprisingly, the remaining density was attributed to mtACP and a small protein L0R8F8, encoded by the upstream open reading frame of the mitochondrial elongation factor 1, MIEF1, also known as AltMIEF1 and AltMid51. L0R8F8 is the bridging protein between MALSU1 and mtACP. L0R8F8 contains the highly conserved LYR motif that mediates its interaction with the mtACP. The function of MALSU1–L0R8F8–mtACP in mitoribosome biogenesis is not entirely clear. However, Brown and colleagues proposed that, during the final stages of assembly when steric obstruction by the MALSU1–L0R8F8–mtACP complex must be released, so that the mt-LSU and mt-SSU can join together, the MALSU1–L0R8F8–mtACP module may prevent premature joining of the two mitoribosomal subunits [47].

The role of L0R8F8 in MALSU1 stabilization and mitoribosome assembly in human cells was established by Dibley and colleagues [210]. Firstly, the AE-MS proteomic studies confirmed the interaction between L0R8F8 and the mtACP (NDUFAB1). Subsequent analysis of topographical proteomic heat maps of the mitoribosome and OXPHOS subunits in cells lacking L0R8F8 were consistent with the BN-PAGE results, suggesting a decrease in mitoribosome and CI and CIV levels. In particular, a reduction in subunits belonging to the mt-LSU and the assembly factor MALSU1 was observed, accompanied by a defect in mitochondrial protein synthesis [210]. These data suggest that L0R8F8 may be required for the stabilization of MALSU1 in order to achieve optimal mitoribosome assembly.

## 7. Concluding Remarks

Localized translation at the mitochondrial inner membrane is closely coupled with nuclear-encoded mitochondrial protein import and the assembly of OXPHOS complexes. Although there are differences between yeast and mammalian coupling of translation and OXPHOS assembly, it is becoming more evident that these are highly coordinated process, with regulation occurring at many different levels, affecting both mitochondrial and cytosolic gene expression, as well as transport machineries and potentially protein quality control systems. In yeast, synchronized translational programs were identified facilitating OXPHOS biogenesis; however, it still remains to be determined whether such coordinated tuning of the two physically separated genomes exists in humans. There is accumulating evidence that downstream regulatory mechanisms, such as import of nuclear-encoded subunits of the OXPHOS system or translational regulators, may enforce mitochondrial translation processes to promote stoichiometric production of OXPHOS subunits. Therefore, it will be important to determine whether (i) mitochondrial import is rate-limiting, such as to control the flux of nuclear-encoded OXPHOS proteins and translational regulators required for translation and OXPHOS biogenesis, or (ii) whether protein quality control pathways can facilitate the biogenesis of dual-origin complexes by regulating protein turnover. Clearly, a great deal remains to be discovered about the mito-nuclear signaling pathways to fully understand the formation of the OXPHOS system.

Although structural studies provided detailed organization of OXPHOS complexes, there remain open questions around the assembly of these multimeric structures. The incorporation of OXPHOS subunits and co-factors is orchestrated via numerous ancillary factors, some of which undergo post-translational modifications, e.g., acylation to facilitate OXPHOS assembly. Indeed, recent studies in yeast hypothesized that modulation of acetyl-coA would lead to changes in the levels of mtACPs, thus affecting the LYRM network of assembly factors required for late-stage OXPHOS assembly. Depending on nutrient availability, this could represent an attractive adaptation model, where the stimulation of OXPHOS assembly is dependent on the metabolic status of the cell linked to acetyl-coA abundance. In humans, such models are largely uninvestigated. However, the high conservation of genes involved opens up new areas of investigation when considering the physiological importance of this model in humans.

## Figures and Tables

**Figure 1 ijms-21-03820-f001:**
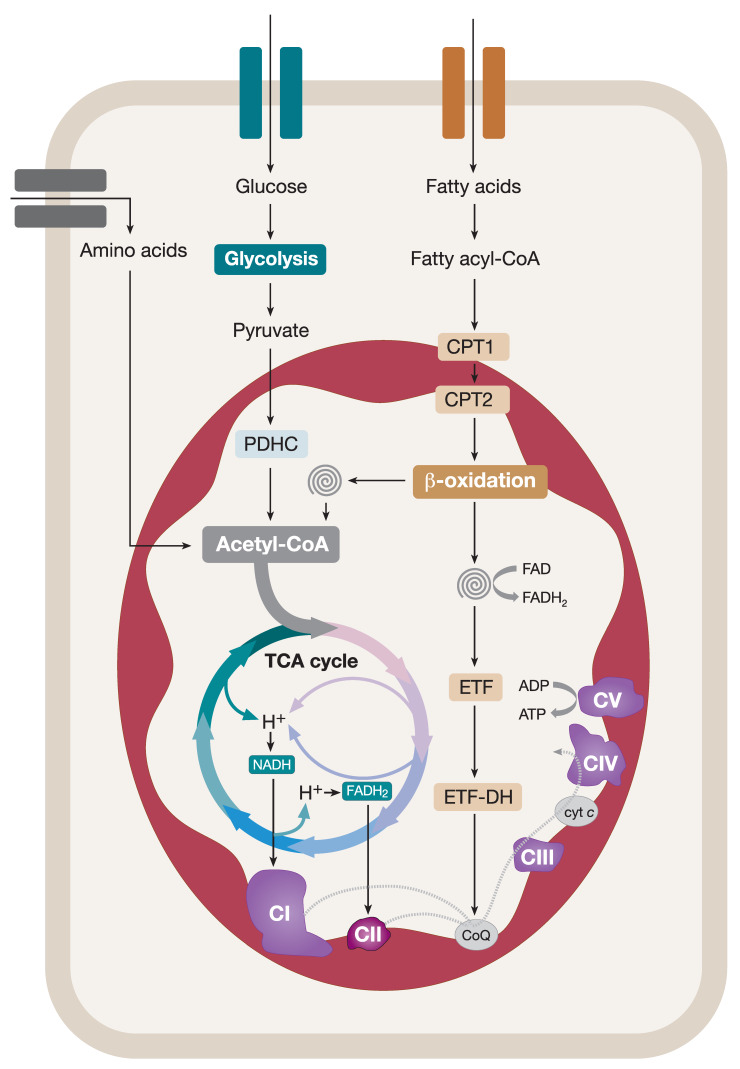
Overview of the main cellular and mitochondrial energy metabolism pathways. Mitochondrial metabolic pathways are fueled by glucose, fatty acids, and amino acids through respective catabolic processes to produce acetyl-CoA, a primary substrate for the TCA cycle, which is also recycled in the fatty acid β-oxidation (FAO) pathway producing NADH and FADH_2_ reducing equivalents that feed the ETC CI and CII, respectively. Glucose undergoes glycolysis in the cytoplasm to form pyruvate which is converted into acetyl-CoA in mitochondria by the PDHC. Acetyl-CoA enters a series of enzyme-catalyzed reactions in the TCA cycle producing reduced forms of NADH and FADH_2_ required for the transport of electrons to the ETC (the flow of electrons is illustrated by a grey dashed line). CI, CIII and CIV pump protons from the matrix to the IMS, generating a proton-motive force across the IMM, that drives the flux of protons back into the matrix via the ATP synthase to harness energy released in the form of ATP. The oxidation of NADH and FADH_2_ via CI and CII replenishes the electron acceptors NAD+ and FAD to maintain TCA cycle activity. Fatty acids are imported into the mitochondria in the form of fatty-acyl-CoA via the carnitine shuttle system (CPT1 and CPT2) to enter the β-oxidation cyclic pathway. Initially, fatty acyl-CoAs undergo dehydrogenation resulting in a production of a FADH_2_ reducing equivalent that transfers electrons to the electron transfer protein ETF and subsequently to the ubiquinone pool via the ETF ubiquinone oxidoreductase (ETF-QO), thus facilitating the entry of electrons to CIII. The remaining three catalytic steps involve hydration, second dehydrogenation during which NAD+ is reduced to NADH that is subsequently re-oxidized by CI, and thiolysis. In the final step, two carbons are removed from the fatty acyl-CoA ester, resulting in shortening of the entry product and its recycling via the FAO pathway. In addition, a molecule of acetyl-CoA is generated that contributes to the overall mitochondrial pool of acetyl-CoA used in the TCA cycle. Amino-acid degradation also contributes to the production of pyruvate, acetyl-CoA, or metabolic intermediates to be oxidized in TCA cycle. Abbreviations are as follows: CI: Complex I, CII: Complex II, CIII: Complex III, CIV: Complex IV, CV: Complex V, CoQ: coenzyme Q/ubiquinone, cyt *c*: cytochrome *c*, Acetyl-CoA: acetyl coenzyme A, PDHC: pyruvate dehydrogenase complex, CPT1/2: carnitine palmitoyltrasnferase 1/2, ETF: electron transfer flavoprotein, ETF-DH: ETF dehydrogenase, NADH: nicotinamide adenine nucleotide, FADH_2_: flavin adenine dinucleotide, ADP: adenosine diphosphate, ATP: adenosine triphosphate.

**Figure 2 ijms-21-03820-f002:**
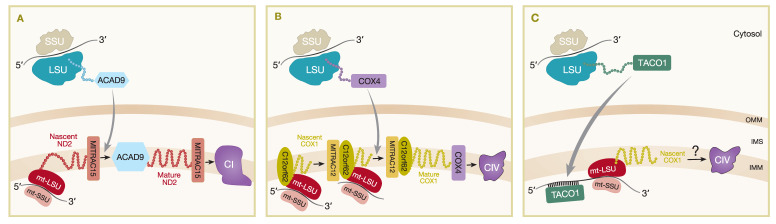
Models of coordination of human mitochondrial and cytosolic translation programs during OXPHOS assembly. Mitochondrial translation exhibits plasticity in coordination with nuclear gene expression by three possible translational regulatory systems identified in humans. (**A**) MITRAC15 binds to *MT-ND2* in a complex with mitoribosome to promote its translation and assembly into CI. ACAD9, a CI assembly factor encoded by the nuclear genome, associates ND2 polypeptides specifically downstream of MITRAC15 for assembly of the ND2 subunit into CI. (**B**) C12orf62 and MITRAC12 act specifically on mitoribosomes translating *MT-CO1* mRNA and as members of the early COX1 assembly intermediates to promote *MT-CO1* translation. The nuclear-encoded COX4 subunit of CIV post-translationally facilitates completion of CIV assembly by binding to the C12orf62/MITRAC12/nascent COX1 protein complex. (**C**) The translational activator TACO1 is hypothesized to bind 5′ end of mtDNA-encoded *MT-CO1* mRNA for the induction of nascent COX1 synthesis to be assembled into CIV. Abbreviations are as follows: SSU: small ribosomal subunit, LSU: large ribosomal subunit, mt-SSU: mitochondrial small ribosomal subunit, mt-LSU: mitochondrial large ribosomal subunit, TACO1: translational activator of cytochrome *c* oxidase I, COX1: cytochrome *c* oxidase subunit 1, COX4: cytochrome *c* oxidase subunit 4, C12orf62: cytochrome *c* oxidase assembly protein COX14, MITRAC12: mitochondrial translation regulation assembly intermediate of cytochrome *c* oxidase protein of 12 kDa, ACAD9: Acyl-CoA dehydrogenase family member 9, MITRAC15: mitochondrial translation regulation assembly intermediate of cytochrome *c* oxidase protein of 15 kDa.

**Figure 3 ijms-21-03820-f003:**
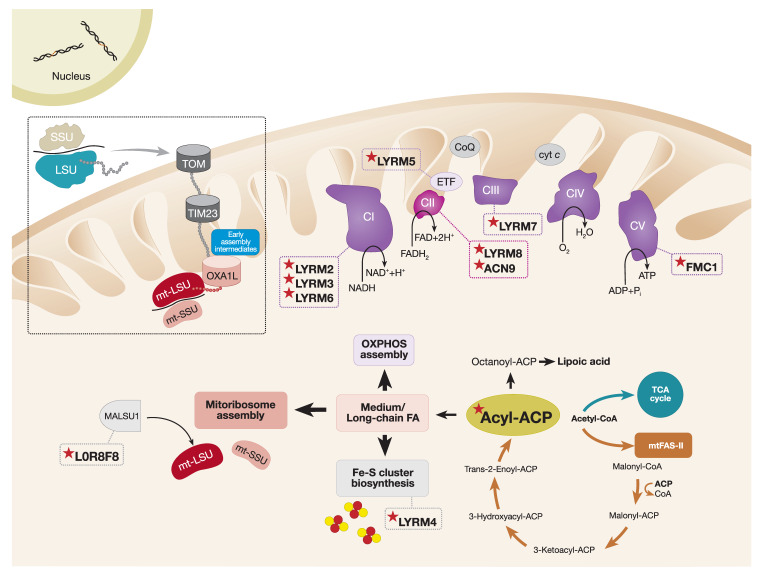
Interactions between the mitochondrial acyl carrier protein and LYRM proteins. The metabolic intermediate acetyl-CoA feeds into two different pathways in the mitochondria: the TCA cycle and mitochondrial type II fatty acid synthesis (mtFAS-II). The mtFAS-II process encompasses stepwise addition of fatty-acyl chain around an ACP scaffold that leads to acylation of ACP. Further elongation of acyl-ACP generates octanoyl-ACP for lipoic acid synthesis. Alternatively, addition of medium- or long-chain fatty acids to acyl-ACP is required for OXPHOS assembly, Fe–S cluster biosynthesis, and mitoribosome assembly through mtACP interaction with LYRM proteins as indicated by the star (★) symbol. The coordination between nuclear and mitochondrial translation required for OXPHOS proteins of dual genetic origin is illustrated within the black dotted box. The majority of nuclear-encoded precursor proteins synthesized by cytosolic ribosomes are imported into the mitochondria by the translocase of the TOM and TIM23 complexes. OXA1L inserts both nuclear- and mitochondrial-encoded proteins into the IMM for the formation of early OXPHOS assembly intermediates (in blue). Abbreviations are as follows: LYRM: leucine–tyrosine–arginine motif, ACN9: acetate non-utilizing protein 9, FMC1: formation of mitochondrial CV assembly factor I homolog, L0R8F8: MIEF1 upstream open reading frame protein, SSU: small ribosomal subunit, LSU: large ribosomal subunit; mt-SSU: mitochondrial small ribosomal subunit, mt-LSU: mitochondrial large ribosomal subunit, MALSU1: mitochondrial assembly of ribosomal large subunit 1, ACP: acyl carrier protein, CoA: coenzyme A, OXA1L: oxidase assembly 1-like protein, TOM: translocase of the outer membrane, TIM23: mitochondrial import inner membrane translocase subunit Tim23.

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
