# Peer review of "Mitochondrial OXPHOS Biogenesis: Co-Regulation of Protein Synthesis, Import, and Assembly Pathways"

_ijms, 2020, doi:10.3390/ijms21113820_

Round 1

Reviewer 1 Report

Tang et al. provide a comprehensive review on the biogenesis of the oxidative phosphorylation system including co-regulation of nuclear and mitochondrial gene expression, mitochondrial protein import, mitochondrial quality control and early and late assembly stages of the OXPHOS complexes. The authors include and discuss relevant recent publications. Hence, this manuscript is suitable for publication in International Journal of Molecular Sciences after consideration of the following issues:

Although the review provides a good overview about recent findings, I get the impression that it is too extensive and the authors might consider to reduce it in some parts. For example, as the main focus is on the OXPHOS biogenesis and its regulation in human/mammals, the authors could cut parts about the yeast system. Additionally, the section about mitochondrial quality control in humans need to be revisited carefully or might be deleted (for details see point 9 and 10).

In the following I have a some suggestions and corrections, which the authors might want to address:

1) line 135: “...3’ polyadenylation, which is essential for the completion of the stop codon,...”

Not all of the transcripts are polyadenylated or require polyadenylation for completion of the stop codon. It would be better to add “...essential for the completion of the stop codon UAA for seven transcripts” (see Temperley et al., 2010)

2) line 143/144: It has been shown that during evolution the mitochondrial ribosome gained first protein mass before reducing its RNA content. Please, rephrase accordingly in the manuscript. (see Van der Sluis et al., 2015)

3) line 146: “...(mtLSU) consisting of 16S rRNA and 53 MRPs.” The mtLSU contains only 52 MRPs (see Greber and Ban, 2016). Amunts et al., 2014 list MRPL56/mL56 with Uniprot ID - P83111 as a MRP, however, it has been shown that this protein encodes for a serine protease involved in mitochondrial lipid metabolism (Keckesova et al., 2017).

4) line 164: “Once mitochondrial translation is terminated, recycling factors will associate at the mitoribosome to release the polypeptide, mRNA and uncharged tRNA...”. This sentence is not correct. The release factor mtRF1a facilitates the hydrolysis of the nascent peptide chain from the peptidyl-tRNA. The substrate for mtRRF and mtEFG2 for ribosome recycling is the mitoribosome with a deacylated tRNA in the P-site. Please, correct.

5) line 189: “Additionally, MALSU1 has been identified as an mt-LSU structural protein ...” This is not correct. MALSU1 is not a structural MRP. It is required for the biogenesis for the mtLSU and has been identified in a late assembly intermediate of the mtLSU by cryo-electronmicroscopy (Brown et al., 2017) as correctly stated in section 6.5. (line 810/811). Please, correct.

6) line 190-192: “The mitoribosomes are anchored to the inner mitochondrial membrane to increase proximity for the insertion of hydrophobic mitochondrial proteins by MRPL45.” This sentence is misleading. The mitoribosome is anchored to the inner membrane by mL45 and the insertion of hydrophobic proteins is facilitated by the insertase OXA1L. Please, correct and use for MRPs the new nomenclature (MRPL45 = mL45).

7) line 271: There is a spelling mistake: “OXA1L invertase”...it should be insertase.

8) line 282: “COX4-1”...I would consistently use COX4 like in figure 2.

9) section 4.2.; line 444-447: There are two types of AAA proteases in human mitochondria - the m-AAA protease and the i-AAA protease. The m-AAA protease exposes its catalytic site to the matrix while the i-AAA protease faces its catalytic site to the intermembrane space. The m-AAA protease is composed of AFG3L2 as homohexamer or of AFG3L2 and SPG7 as a heterohexamer. The i-AAA protease is formed by YME1L. As written in the manuscript it sounds that the AAA protease composed of AFG3L2 and SPG7 can face the matrix as well as the intermembrane space. Please, correct.

10) line 453-455: “Battersby and colleagues have proposed that the disruption to the mitochondrial AAA proteolytic quality control pathway induces the activation of OMA1 which cleaves the IMM-anchored GTPase, OPA1...” It has been already shown by Ehses et al. (2009) that defects in the m-AAA protease induce OMA1 activation and therefore OPA1 cleavage. Please, cite correctly.

Additionally, the human mitochondrial quality control includes more than just the AAA proteases.

I am not sure whether the section of mitochondrial quality control is really necessary in the context of this manuscript. If the authors want to include quality control in human mitochondria the section requires corrections and more recent published literature. It is not true that there are limited studies in human cells regarding mitochondrial quality control (line 439-440). Please, see recent studies by the groups of Thomas Langer and Elena Rugarli.

10) line 602/603: “It was proposed that COA6 and COX16 also play important roles in the assembly of CuA center, but the underlying mechanism is still unknown.” It has been recently shown that COA6 acts as thiol-reductase reducing disulfide bridges in SCO1 and SCO2, essential for CuA biogenesis (see Pacheu-Grau et al., 2020).

11) line 629-630: “The oligomerization of c subunits to form the rotor c-ring is possibly facilitated by TMEM70 to assemble...” Recent studies also link TMEM70 to CI assembly (see Sànchez-Caballero et al., 2020).

12) section 6.1. vs figure 3 vs section 6.5.: It would be better if the author consistently use one abbreviation for AltMid51/L0R8F8. In section 6.1. they use AltMid51, but in figure 3 and section 6.5. they use L0R8F8.

13) Figure 3: In my opinion it is a strange arrangement of the mitoribosome, OXA1L and CI with the nascent peptide chain, which seems to be an elongation of the mt-mRNA. I would suggest to remove the cristae in this figure as it would make it easier to arrange all the complexes and components.

The mitoribosome is facing the inner membrane with the mtLSU, not with the mtSSU. Please, correct.

Author Response

Please see attached file that addresses Reviewer 1 comments.

Reviewer 2 Report

The manuscript of Tang et al., overview the literature on the very exiting question of mitochondrial gene expression, mitochondrial protein synthesis, and assembly of components of electron transport chain (ETC). The theme of the review is very interesting and the authors did a very good job of collecting and analyzing published data.
However, I have some concerns about the manuscript.
The major one is that the introduction section is heavy for the reader. Oxidative Phosphorylation described in a confusing manner. Moreover, the authors described carefully all the steps of energy metabolism in the introduction and in the legend of figure 1 but did not mention the key points, which are proton transport through the inner membrane and created proton gradient.
The manuscript is full of abbreviations of proteins, some of them appear without a transcript and without any brief background of protein functions (e. g. MRPL45, MITRAC complexes, Vms1, etc.).
The manuscript describes a lot of complex mechanisms of regulation of mitochondrion biogenesis, transportation, and incorporation of proteins inside the inner mitochondrial membrane, assembly of complexes of the, BUT absolutely lack of figures. Only 3 illustrations are presented in 20 pages of text. More illustrations would make the material more clear.
Overall, I like the idea and the material summarized in the manuscript. I believe that the improvement of the points mentioned above would increase the quality of the review.

Author Response

Please see attached file that addresses Reviewer 2 comments.

Reviewer 3 Report

I found this review very well organized, complete and at the same time concise.

I have only some comments for the authors to improve the current manuscript to make it more readable and thorough:

I think that this review lacks adequate references to pathologies related to the pathways described. I don't ask to list all the disease related to mutations of the listed genes, but I think that giving some examples of disease during the discussion, or inserting a table with the most relevant diseases connected to these pathways would be useful.

For the part 4.1. Mitochondrial quality control pathways in yeast: authors should expand the discussion describing the physiological impact of peptide turnover on mitochondrial function.

Upon import, presequences are typically cleaved by the mitochondrial processing peptidase MPP in the matrix releasing the mature protein. In several cases, MPP generates import intermediates that are further processed by the octapeptidyl peptidase Oct1/MIP or the intermediate cleaving peptidase Icp55. This is a very important step, because incomplete processing of mitochondrial preproteins due to MPP mutations or dysfunction leads to their destabilization and accelerated turnover.

Presequence peptides that have been cleaved by MPP are subsequently degraded by the matrix peptidasome Cym1/Ste23. A problem in this step impairs MTS degradation and a block of the MPP activity.

I suggest to review two key papers  about this topic:

Mossmann D, et al. Amyloid-β peptide induces mitochondrial dysfunction by inhibition of preprotein maturation.Cell Metab. 2014 Oct 7; 20(4):662-9.

Asli Aras Taskin, et al. The novel mitochondrial matrix protease Ste23 is required for efficient presequence degradation and processing Mol Biol Cell. 2017 Apr 15; 28(8): 997–1002. doi: 10.1091/mbc.E16-10-0732

For the part 4.2. Mitochondrial quality control pathways in humans:

as mentioned above, it should be useful to mention the role of PITRM1, the cym1 analogue. Impairment of Pitrm1 leads to MTS accumulation and induec a negative feedback on MPP, with conseguent block of imported proteins maturation. Moreover mutations in this gene are related to a neurodegenerative disease.

Another important point to mention may be this recently finding:

Respiratory supercomplexes act as a platform for complex III-mediated maturation of human mitochondrial complexes I and IV.

Protasoni M, et al. EMBO J. 2020 Feb 3;39(3):e102817. doi: 10.15252/embj.2019102817. Epub 2020 Jan 8.

5.4. Complex IV (cytochrome c oxidase) assembly: in this section, the role of Surf1 should be explained better. Thsi is a very important protein for the CIV assembly.

Another important point to mention should be the link between Opa1 and the oxphos.

Minor points:

line 67, space missed in  FADH2reducing

line 77: ETF:ubiquinone, remove : and use space

line 178-180: Bogenhagen and colleagues modelled the assembly of the mitoribosome by utilizing pulse SILAC (Stable Isotope Labelling by Amino Acids in Cell Culture) of MRPs in HeLa cells [133].  133 is not the right  reference

Quality of the figure should be improved, expecially figure 2

Author Response

Please see attached file that addresses Reviewer 3 comments.
